# A ^13^CO_2_ Enrichment Experiment to Study the Synthesis Pathways of Polyunsaturated Fatty Acids of the Haptophyte *Tisochrysis lutea*

**DOI:** 10.3390/md20010022

**Published:** 2021-12-24

**Authors:** Marine Remize, Frédéric Planchon, Matthieu Garnier, Ai Ning Loh, Fabienne Le Grand, Antoine Bideau, Christophe Lambert, Rudolph Corvaisier, Aswani Volety, Philippe Soudant

**Affiliations:** 1UMR 6539 LEMAR, CNRS, IRD, Ifremer, University of Brest, 29280 Plouzane, France; Frederic.Planchon@univ-brest.fr (F.P.); fabienne.legrand@univ-brest.fr (F.L.G.); Antoine.Bideau@univ-brest.fr (A.B.); christophe.lambert@univ-brest.fr (C.L.); rudolph.corvaisier@univ-brest.fr (R.C.); 2GREENSEA, Promenade du Sergeant Navarro, 34140 Meze, France; 3PBA, Ifremer, Rue de l’Ile d’Yeu, BP 21105, CEDEX 03, 44311 Nantes, France; Matthieu.Garnier@ifremer.fr; 4Center for Marine Science, Department of Earth and Ocean Sciences, University of North Carolina Wilmington, 5600 Marvin K. Moss Ln, Wilmington, NC 28403, USA; lohan@uncw.edu; 550 Campus Drive, Elon University, Elon, NC 27244, USA; avolety@elon.edu

**Keywords:** long-chain PUFA synthesis, desaturases, elongases, PKS pathway, 20:5n-3 (EPA), 22:6n-3 (DHA), *Tisochrysis lutea*, ^13^C artificial enrichment

## Abstract

The production of polyunsaturated fatty acids (PUFA) in *Tisochrysis lutea* was studied using the gradual incorporation of a ^13^C-enriched isotopic marker, ^13^CO_2_, for 24 h during the exponential growth of the algae. The ^13^C enrichment of eleven fatty acids was followed to understand the synthetic pathways the most likely to form the essential polyunsaturated fatty acids 20:5n-3 (EPA) and 22:6n-3 (DHA) in *T. lutea*. The fatty acids 16:0, 18:1n-9 + 18:3n-3, 18:2n-6, and 22:5n-6 were the most enriched in ^13^C. On the contrary, 18:4n-3 and 18:5n-3 were the least enriched in ^13^C after long chain polyunsaturated fatty acids such as 20:5n-3 or 22:5n-3. The algae appeared to use different routes in parallel to form its polyunsaturated fatty acids. The use of the PKS pathway was hypothesized for polyunsaturated fatty acids with n-6 configuration (such as 22:5n-6) but might also exist for n-3 PUFA (especially 20:5n-3). With regard to the conventional n-3 PUFA pathway, Δ6 desaturation of 18:3n-3 appeared to be the most limiting step for *T. lutea*, “stopping” at the synthesis of 18:4n-3 and 18:5n-3. These two fatty acids were hypothesized to not undergo any further reaction of elongation and desaturation after being formed and were therefore considered “end-products”. To circumvent this limiting synthetic route, *Tisochrysis lutea* seemed to have developed an alternative route via Δ8 desaturation to produce longer chain fatty acids such as 20:5n-3 and 22:5n-3. 22:6n-3 presented a lower enrichment and appeared to be produced by a combination of different pathways: the conventional n-3 PUFA pathway by desaturation of 22:5n-3, the alternative route of ω-3 desaturase using 22:5n-6 as precursor, and possibly the PKS pathway. In this study, PKS synthesis looked particularly effective for producing long chain polyunsaturated fatty acids. The rate of enrichment of these compounds hypothetically synthesized by PKS is remarkably fast, making undetectable the ^13^C incorporation into their precursors. Finally, we identified a protein cluster gathering PKS sequences of proteins that are hypothesized allowing n-3 PUFA synthesis.

## 1. Introduction

Long chain polyunsaturated fatty acids (LC-PUFA) such as 20:5n-3 (EPA) and 22:6n-3 (DHA) are important compounds for most marine metazoans for their growth, reproduction, and development. They are not able to synthetize them in sufficient quantities and thus have to acquire them from their diet. On the basis of the food web, protists are the main producers of these fatty acids and present a key role in marine ecosystem functioning. 20:5n-3 and 22:6n-3 are also particularly important in human nutrition. They are known to have beneficial effects on cardiovascular diseases or diabetes. However, due to high demand for human nutrition and aquaculture of carnivore species, a shortage of these two compounds found in fish oil is predicted to occur by 2050 [1]. Despite their economic and ecologic interests, biological and ecological processes responsible for their synthesis are still under investigation. It is, then, of first concern to understand how 20:5n-3 and 22:6n-3 are produced at the basis of the food webs, and how global changes could affect their availability at higher trophic levels.

In phytoplankton and microzooplankton, fatty acids are synthetized via different metabolic pathways [2,3,4]. The most “conventional” pathway is the fatty acid synthase (FAS) pathway, followed by the elongation and front-end desaturation steps of the n-3 and n-6 pathways. Starting with the initial formation of acetyl-CoA and then malonyl-CoA in aerobic conditions, these pathways produce more complex fatty acids by progressive addition of two atoms of carbon (elongation steps) or desaturations of precursors such as 16:0 or 18:0 [5,6,7]. These two pathways can be connected by the so-called ω-3 desaturase (or methyl end desaturase) pathway. Within the n-3 and n-6 pathways, an alternative route of Δ8 desaturation can also bypass the Δ6 desaturation step and has already been identified in Haptophyte [8]. These routes allowed the synthesis of 20:5n-3 as well as 22:6n-3 (Figure 1).

An alternative O_2_ independent pathway, called the polyketide synthase (PKS) pathway, is responsible of long chain PUFA synthesis such as 20:5n-3 and 22:6n-3 [3,9,10]. It has been found in bacteria and protists such as thraustochytrids, dinophytes, and haptophytes [11,12,13,14,15]. The PKS pathway relies on the same four basic enzymatic reactions (condensation, reduction, dehydration, and reduction) as the FAS pathway. Opposed to the conventional pathway, the PKS pathway is less energy consuming, because it requires fewer reduction and dehydration steps than “conventional” pathways [3]. The metabolites used to form the carbon chain are simultaneously desaturated and elongated, creating long-chain PUFA [3,16,17].

Even if some microalgae species share all or part of O_2_-dependent n-3 and n-6 pathways and O_2_-independent PKS pathways, PUFA composition of primary producers varies greatly according to species. Diatoms synthetize more 20:5n-3 as well as C16 PUFA, while dinophytes or haptophytes contain more 22:6n-3 or C_18_ PUFA. Other groups such as cyanobacteria or some chlorophytes classes are unable to build 20:5n-3 or 22:6n-3 or only in very low proportions (<1%) [18,19,20,21].

To improve knowledge of the synthesis routes and production of 20:5n-3 and 22:6n-3, studies have focused on identifying genes coding for the different elongase and desaturase [10,22,23,24,25]. Moreover, in recent years, the use of ^13^C substrate allowed monitoring the incorporation of labelled substrates into targeted organic macromolecules. Different metabolic intermediates or end products such as fatty acids [26,27,28,29] are monitored and quantified. This has already been applied to *E. coli* [30], yeast [29], and microalgae [31,32,33,34,35]. The development of technics such as gas chromatography coupled to mass spectrometry (GC-c-IRMS) consists of a noticeable improvement in the direct resolution of isotopic composition of organic macromolecules the so-called compounds specific isotope analysis (CSIA) including fatty acids [36,37,38,39].

The present study aims at investigating the synthesis pathways of essential PUFA 20:5n-3 and 22:6n-3 using stable isotope (^13^C) labelling experiment of the haptophyte *Tisochrysis lutea*. *T. lutea* is intensively used in aquaculture (hatchery) and industry [40]. The incorporation of ^13^C was monitored in 11 FA during 24 h at a high temporal resolution (each 0.5 to 2 h). Progressive accretion of the ^13^C-labelled CO_2_ into FA (from precursors to PUFA of interest) allowed us to constrain FAS, elongase/desaturase, and PKS involvement in 20:5n-3 and 22:6n-3 production by *Tisochrysis lutea*. In parallel to the monitoring of ^13^C incorporation into FA, growth, physiological status, and other cellular parameters (morphology, viability, esterase activity, and lipid content) were monitored by flow cytometry analysis.

## 2. Results

### 2.1. Algae Physiology and Biochemistry during the 24 h Experiment

Cell abundance for *Tisochrysis lutea* during the 24 h of experiment increased sharply from t_0_ to t_24_. The experiment allowed cell concentration to double for the three balloons (Figure 2A). Despite the attention given to homogenization at the time of subculture from inoculum, the second enriched balloon had a cell concentration twice higher than Tl1 and TlT. This difference remained constant during the entire experiment. Cell abundance varied from 4.3 × 10^6^ cells·mL^−1^ to 9.5 × 10^6^ cells·mL^−1^ for the most concentrated balloon TI2 and from, on average, 2.6 × 10^6^ cells·mL^−1^ to 6.4 × 10^6^ cells·mL^−1^ for the two others (Figure 2A). Despite the concentration differences, the general slopes for the three balloons were very similar (0.22 cells·mL^−1^·h^−1^ for Tl2 and 0.15 cells·mL^−1^·h^−1^ for Tl1 + TlT) (Figure 2A). Bacteria were also found in higher abundance Tl2 (Figure 2B), almost five times higher than in Tl1 and TlT. However, bacteria increased only by a factor of 1.2 for Tl2 between t_0_ and t_24_ versus a factor 3.5 in average for Tl1 and TlT. Bacteria concentration was around 6.6 times higher than algae concentration for Tl1 and TlT on average over the 24 h of the experiment. For Tl2, bacteria concentration was 16 times higher than algae concentration at the beginning of the experiment, but this ratio decreased progressively until t_24_ (8.7 times higher) (Figure 2B).

FSC and SSC were, respectively, considered a proxy of cell size and cell complexity, and FL3 was considered a proxy of chlorophyll content. SSC and red fluorescence (FL3) did not significantly vary during the entire experiment (Bartlett tests, *p* > 0.05 and ANOVA *p* > 0.05) (Appendix A). FSC increased slightly with time for the three balloons (Bartlett test, *p* > 0.05 and ANOVA *p* = 0.03) (Appendix A). The percentage of dead microalgae (as measured by SYTOX staining assay) remained below 7% for the 24h of the experiment (Appendix A).

Particulate organic carbon concentration increased similarly to the cell abundance for the two labelled balloons from 3.6 to 9.0 mmolC·L^−1^ for Tl1, from 4.5 to 11.6 mmolC·L^−1^ for Tl2, and from 3.4 to 8.2 mmolC·L^−1^ for TlT (Figure 3A). Increase in POC, as a function of experiment duration (*R*^2^ = 0.81, *p* < 0.0001), occurred at a relatively constant rate of 0.2 mmolC·L^−1^·h^−1^ for Tl1, Tl2, and TlT considered together. Total fatty acid (TFA) concentration also increased for the three balloons (between 0.26 mmolC·L^−1^ to 0.60 mmolC·L^−1^ for Tl1, from 0.25 mmolC·L^−1^ to 0.80 mmolC·L^−1^ for Tl2, and from 0.17 mmolC·L^−1^ to 0.55 mmolC·L^−1^ for TlT) (Figure 3B). The slope was 15 µmolC·L^−1^·h^−1^ (*R*^2^ = 0.72, *p* < 0.0001). POC was significantly correlated to cell concentration (*R*^2^ = 0.80 *p* < 0.0001). The slope of the relation between POC and cell abundance is a proxy of carbon content per cell for *T. lutea,* which was, on average for the three balloons, equal to 1.07 fmolC·cell^−1^ (Figure 3C). TFA concentration was linearly and positively correlated with POC concentration (*R*^2^ = 0.73, *p* < 0.0001). The slope of the regression between TFA and POC concentration indicates that TFA represent in average 7.7% of bulk POC (Figure 3D).

### 2.2. ^13^C Atomic Enrichment (AE) of Particulate Organic Carbon and Dissolved Inorganic Carbon

Dissolved inorganic carbon (DIC) progressively enriched in the two balloons with ^13^CO_2_ (Figure 4A). The enrichment trends were similar for the two balloons after t_4_, with an important increase in the DIC atomic enrichment until t_20_ (up to 58.1 and 61.1% for Tl1 and Tl2, respectively). The increase in AE tended to stabilize after t_20_. Final levels of enrichment were 61.5 and 64.6% for Tl1 and Tl2, respectively (Figure 4A). Atomic enrichment (AE_POC_) increased sharply after t_4_ for the two balloons until the end of the experiment. Enrichment levels at t_24_ were 25.2 and 34.7% for Tl1 and Tl2, respectively (Figure 4B).

### 2.3. Fatty Acid Composition in Neutral and Polar Lipids in T. lutea

Neutral lipids and polar lipids represented, respectively, 37% and 63% of TFA on average for the three balloons (Figure 5A). The proportions of individual fatty acid in NL and PL did not vary throughout the experiment. Total bacteria fatty acids (iso15:0, ante15:0, iso16:0, iso17:0, 15:0, 17:0, 21:0, 15:1n-5—Appendix A) remained below 1% for both NL and PL fractions during the 24 h. Branched fatty acids were only present in trace amounts (Appendix A). Concentrations in µg·L^−1^ and µmolC·L^−1^ as well as proportions in% of all identified and quantified FA in neutral and polar lipid fractions according to sampling time are available in the Appendix A.

We focus the presentation of the results on the polar lipid fraction, as it is the predominant fraction containing FA (Figure 5B). During the experiment, thirty two fatty acids (FA), as listed in the Material and Methods section, were identified and quantified for *T. lutea*, with 12 being over 1% of the TFA in PL (14:0, 16:0, 18:0, 16:1n-7, 18:1n-9, 18:1n-7, 18:2n-6, 18:3n-3, 18:4n-3, 18:5n-3, 22:5n-6, and 22:6n-3) (Figure 5A). Although under 1% for PL, the 16:3n-6, 20:5n-3, and 22:5n-3 were also presented due to their potential synthesis significance (Figure 5A). PUFA (in average 30%) and SFA (21%) were the main FA categories for polar lipids (PL) during the 24 h. PUFA n-3 represented 25% of the TFA, PUFA n-6 5%.

In PL, 14:0 and 22:6n-3 (respectively, 21% and 18% on average over the 24 h) were the most abundant, followed by 18:1n-9, 18:4n-3, and 16:0 (11–13% of TFA). Finally, 18:3n-3, 18:5n-3, and 22:5n-6 ranged from 3 to 4% of the TFA (Figure 5B). Patterns observed for NL are available in Appendix A.

### 2.4. Fatty Acid ^13^C Atomic Enrichment

Figure 6 shows the atomic enrichment (AE) of the eleven fatty acids over time. Despite the different timing and level of enrichment between the two balloons, the temporal dynamic of fatty acids enrichment remained similar. The 18:2n-6 and 18:1n-9 + 18:3n-3 had the highest AE during the entire experiment. 22:5n-6, 16:0, and finally, 20:5n-3 were next. The less enriched fatty acids in the polar lipid fraction were in decreasing order 22:5n-3, 22:6n-3, 18:4n-3, 18:5n-3, and finally, 18:0 (Figure 6). For the NL (Appendix A), 20:5n-3, 22:5n-6, 18:1n-9, 16:0, and 18:2n-6 were always the most enriched. The sequence for the other fatty acids remained close to that of polar lipids. It has to be noted that enrichments of 20:5n-3 and, to a lesser extent, of 22:5n-3 were higher in NL than in PL.

Table 1 explored FA synthesis pathways with regard to their most expected direct precursor. Most ratios were below 1, except for the 20:5n-3/18:5n-3 ratio, which was above 1. Similar patterns were observed in NL (Appendix A).

### 2.5. Identification of Candidate Proteins for PKS Synthesis in T. lutea

Thirty sequences of potential candidate proteins involved in *T. lutea* PUFA synthesis have been identified and are presented in Appendix A. Only fourteen presented the four main domains potentially coding for the enzymes used in PKS PUFA synthesis pathways: ketoacyl reductase (KR), polyketide synthase (KS), dehydrase/dehydrogenase (DH), and enoyl reductase (ER). Among these sequences, four sequences (TISO_14962, TISO_14968, TISO_14975, and TISO_14977) were part of the same cluster (group of homologous proteins) and presented multiple KS, KR, ER, and DH domains as well as phosphopantetheine (PP)-binding domains (Figure 7). TISO_14962 also possessed methyltransferases and thioesterase domains (Figure 7). TISO_14977 presented a domain acknowledged to be involved in acetyl-CoA synthesis. Within this cluster, TISO_14973 was also selected, as it contains an atypical domain, specifically recognized as being involved in n-3 PUFA synthesis. Nine other sequences (TISO_04539, TISO_06404, TISO_06537, TISO_08047, TISO_11097 TISO_16495, TISO_27353, TISO_37260, and TISO_37631) were also found, containing the four main domains (up to 18 for KR in TISO_08047). Except TISO_37631, these sequences also have thioesterase, sulfotransferase, or peptide-synthesis-related domains, and thus they might be in charge of the synthesis of more complex lipids.

## 3. Discussion

This study investigated long-chain PUFA synthesis pathways in the haptophyte *Tisochrysis lutea* using the incorporation of ^13^CO_2_. Addition of ^13^CO_2_ did not affect *T. lutea* physiology. Cell viability remained above 93% during the experiment, while cell complexity and chlorophyll content did not vary significantly according to sampling time. Cell size (as attested by FSC) increased slightly during the 24 h experiment. *T. lutea* produced FA to a level of 7% of POC; predominantly in the form of PL (66%).

Major FA of *T. lutea* were similar in proportions to those found in other prymnesiophycea (Haptophytes), i.e., 14:0, 16:0, 18:1n-9, 18:4n-3, and 22:6n-3 [41,42,43,44,45]. As reported before in Huang et al. (2019) [46], *T. lutea* had a low content of neutral lipids during exponential phase, and PUFA were mainly found in the polar fraction. *Tisochrysis lutea* accumulates neutral lipids mainly during stationary phase or under nutritive limitations [45].

The final level of atomic enrichment (AE) into the different FA witnessed active synthesis, as most fatty acids had a higher AE than that of POC (30% on average for the two balloons). 22:5n-6 was the most enriched long chain PUFA (LC-PUFA) in the PL fraction. 22:5n-6 and 18:2n-6 were the only ^13^C labelled n-6 fatty acids detectable by GC-c-IRMS. None of the known synthesis intermediates (18:3n-6, 20:3n-6, 20:4n-6, and 22:4n-6) between 18:2n-6 and 22:5n-6 [4] had measurable ^13^C-labelling and were below 1% in the FA profile during our experiment. It is then difficult to hypothesize the pathway used to create 22:5n-6 with this missing information. However, even though the different intermediates were undetectable, 18:2n-6 and 22:5n-6 atomic enrichments being very close cannot exclude them to be related to each other. While studying the existence of an alternative Δ8 desaturase in Haptophyte, Qi et al. (2002) [8] noticed the absence of intermediates of the n-6 Δ8 desaturase pathway (20:2n-6, 20:3n-6 and 20:4n-6)) in *Isochrysis galbana*. It was attributed to relatively high active enzymes that could form the end-product 22:5n-6 with a rapid flow through these n-6 intermediates. Our results agree with this, as ^13^C enrichment of n-6 intermediates could not be detected by compound specific isotope analysis. To demonstrate the existence of these pathways, it would be interesting to combine functional analysis of desaturases by expression in yeast and GC-c-IRMS monitoring of the intermediates after ^13^C labelling of their precursors.

However, it is also possible that another pathway not involving “classical” n-6 FA intermediates exist in *T. lutea*. Previous studies showed the existence of PKS genes in various species of the prymnesiophytes including *Isochrysis galbana* [47], closely phylogenetically related to *Tisochrysis lutea*. We identified five candidates; proteins potentially involved in PKS synthesis pathway in *T. lutea*. Even if their function has not been verified, it is possible that at least one of the proteins presented in Figure 7 was responsible for the formation of n-6 PUFA in the haptophyte. Thus, our hypothesis is that an n-6 PKS pathway might also exist in *T. lutea* (Figure 8). Finally, PKS and “classical” n-6 routes might not be completely independent and could interact in the synthesis of 22:5n-6 in *T. lutea*.

Despite being one of the most abundant FA, 18:4n-3 showed a low ^13^C-enrichment (23%). The synthesis of 18:4n-3 from 18:3n-3 by Δ6 desaturase had already been described by *Isochrysis* sp. [48]. We assume that such activity also exists in *Tisochrysis*, phylogenetically close to *Isochrysis*. However, as 18:3n-3 co-elute with 18:1n-9, it was not possible to measure its AE and to assess whether this could be a limiting step in n-3 pathway (Figure 9). The 18:5n-3 had the lowest enrichment, and the ratio 18:5n-3/18:4n-3 was below the threshold value (R = 0.78), indicating a feasible transformation of 18:4n-3 into 18:5n-3. The existence of Δ3 desaturase that could support the production of 18:5n-3 (18:5Δ3,6,9,12,15) from 18:4n-3 (18:4Δ6,9,12,15) had been suggested by Joseph (1975) [49] to explain the presence of this unusual FA in dinophytes. A more recent study by Ahman et al. (2011) [23] showed in *Ostreococcus lucimarinus* that a Δ4 desaturase was surprisingly able to add a double bond in 18:4n-3 at the Δ3 position leading to the formation of 18:5n-3 when the gene was expressed in yeast cell and supplemented by 18:4n-3 as substrate. With our results and the discovery of Ahman et al. (2011) [23], we proposed that a Δ4 desaturase of *T. lutea* might be able to act as a Δ3 desaturase on 18:4n-3 to produce 18:5n-3 (Figure 9). Desaturation of 18:4n-3 into 18:5n-3 had been previously hypothesized by Kotajima et al. (2014) [50] in the prymnesiophyte *Emiliania huxleyi*.

The 18:5n-3 was also described as an intermediate of 22:6n-3 synthesis by PKS pathway [4]. However, its low enrichment, as compared to 22:6n-3, appeared not compatible with a hypothetical production through this pathway. Nevertheless, one may speculate that there are two separated PKS pathways, one for the 22:6n-3 and one for the 18:5n-3, as these two PUFA are localized in different cell compartments. The 18:5n-3 is generally associated with chloroplastic glycolipids, while the 22:6n-3 is predominant in the other cellular compartments [51,52,53].

Surprisingly, 20:5n-3 in PL was more enriched than 18:4n-3, its precursor in the n-3 pathway [4]. As AE of 20:5n-3 is higher than AE of 18:4n-3, it seems very unlikely that 20:5n-3 was produced via the pathway involving 18:4n-3 elongation and 20:4n-3 Δ5 desaturation. The existence of the alternative Δ8 desaturase pathway have been studied before in *Isochrysis galbana* and *Pavlova lutheri* [8,54,55]. However, as for the n-6 PUFA, intermediates (20:3n-3 and 20:4n-3) of the alternative Δ8 pathway were not detected by fatty acid analysis of *Isochrysis galbana* [8]. Similarly, in our study, intermediates (20:3n-3 and 20:4n-3) of this pathway to synthesize 20:5n-3 have not been found in sufficient amount to be measured by CSIA. As proposed by Qi et al. (2002) [8] for *Isochrysis galbana*, the synthesis of 20:5n-3 via 20:3n-3 and 20:4n-3 by *Tisochrysis lutea* might be very rapid, explaining why these two intermediates were only found in trace amounts (0.12% and 0.02% in PL and 0.05% and 0.33% in NL, respectively).

Due to their lower enrichments, 18:4n-3 and 18:5n-3 seemed unlikely involved in long chain PUFA synthesis such as 20:5n-3 and 22:6n-3. Based on the enrichment dynamics, elongation of 20:5n-3 into 22:5n-3 and further desaturation into 22:6n-3, respectively, by Δ5 elongase and Δ4 desaturase could be possible in *Tisochrysis lutea*. Ratio 22:5n-3/20:5n-3 in PL was within the threshold, indicating a simultaneous enrichment of both 20:5n-3 and 22:5n-3 in *T. lutea*. Such enzymes have been evidenced in haptophytes [54,56,57].

Considering the diversity of PKS gene in haptophytes [47], the possibility of production of 22:6n-3 directly by PKS PUFA synthesis pathway might be possible, as previously shown with thraustochytrids [10,58]. Synthesis of 22:6n-3 by PKS pathway might be at play in parallel with the n-3 pathway. Indeed, we identified a protein cluster gathering the four main domains potentially coding for the enzymes used in PKS PUFA synthesis pathways: ketoacyl reductase (KR), polyketide synthase (KS), dehydrase/dehydrogenase (DH), and enoyl reductase (ER). Protein clusters are groups of similar proteins that most likely shared the same or similar functions [59]. By considering this cluster (candidate proteins TISO_14962, TISO_14968, TISO_14968, TISO_14973, TISO_14975, and TISO_14977, Figure 8), it could be possible that these proteins act together and allow n-3 PUFA synthesis via PKS pathway. Interestingly, protein TISO_14973, while possessing only two of the four domains of interest (KS and DH), presented a specific n-3 domain. This protein might act concomitantly with the other proteins of the same cluster and allow the access to the missing reductase activities (KR and ER). Finally, the ratio 22:5n-6/22:6n-3 was below the threshold value making possible the conversion of 22:5n-6 into 22:6n-3 if we assumed that ω3-desaturase might exist in haptophyte. Synthesis of 22:6n-3 by both n-3 and n-6 pathway might be feasible in *Tisochrysis lutea* (Figure 10). These different ways to produce 22:6n-3 might contribute to betaine lipids synthesis. Indeed, betaine lipids are generally highly unsaturated in C_20_ and C_22_ PUFA, especially in 22:6n-3 in haptophytes [60,61,62,63].

## 4. Material and Methods

### 4.1. Algal Culture and ^13^C Labelling

This study was conducted following the experimental design described by Remize et al. (2020) [34]. The marine prymnesiophyte *Tisochrysis lutea* (T-iso, CCAP 927/14) was cultured in 2 L batch condition in balloons under continuous light (24 h light cycle, 100 µmoles photons m^−2^·s^−1^) at 20 °C and with pH regulation at 7.50 ± 0.05 by CO_2_ injection. Filtered seawater was previously enriched with Conway medium [64] and inoculated with *T. lutea* preculture in a growing stage (exponential phase, four days old). The experimental setup was composed of two cultures (Tl1 and Tl2) receiving the labelling ^13^C-CO_2_ gas (Sigma-Aldrich, <3%atom ^18^O, 99.0%atom ^13^C) and one culture (TlT) receiving petrochemical CO_2_ gas. ^13^C-incorporation in Tl1 and Tl2 began after inoculation (t_0_) and was maintained for 24 h (t_24_).

During the first hours of the experiment, the ^13^CO_2_ injection tube of balloon TI1 had been temporarily disconnected from the system. Consequently, balloon Tl2 received earlier the ^13^CO_2_ and thus started to incorporate ^13^C before balloon Tl1.

### 4.2. Samples Collection

Sampling was performed as described in Remize et al. (2020) [34], i.e., at 30 min, 1 h, 2 h, 3 h, 4 h, and then every 2 h. A total of 16 samples was collected during the 24 h monitoring. At each sampling time, a total volume of 30 to 70 mL was collected for (i) flow cytometry analysis of cellular parameters, (ii) bulk isotopic analysis of particulate organic carbon (^13^C-POC) and dissolved inorganic carbon (^13^C-DIC) by EA-IRMS, (iii) fatty acid (FA) analysis in neutral lipids (NL) and polar lipids (PL) by GC-FID, and (iv) compound specific isotope analysis (CSIA) of FA (^13^C-FA) by GC-IRMS, as described in the following paragraphs.

### 4.3. Flow Cytometry Analysis

Algae growth cellular variables were measured using an Easy-Cyte Plus 6HT flow cytometer (Guava Merck Millipore^®^, Darmstadt, Germany) equipped with a 488 nm blue laser, detectors of forward (FSC) and side (SSC) light scatters, and three fluorescence detectors: green (525/30 nm), yellow (583/26 nm), and red (680/30 nm). The protocol, the variables studied, and the probes used for this cytometry following are described in Remize et al. (2020) [34]. Briefly, forward scatter (FSC), side scatter (SSC), and red fluorescence (FL3, red emission filter long pass, 670 nm) are used to study, respectively, cell size, complexity, and chlorophyll content. The fluorescent probe (SYTOX, Molecular Probes, Invitrogen, Eugene OR, USA, final concentration of 0.05 µM) was used to assess cell viability on FL1 detector (green fluorescence). The concentration of bacteria was also monitored by using SYBR^®^Green (Molecular Probes, Invitrogen, Eugene, OR, USA, #S7563) on FL1 detector. Concentrations of algae and bacteria were given cells per mL, and cellular variables were expressed in arbitrary units (a.u).

### 4.4. POC Concentration and Bulk Carbon Isotopic Composition

For particulate organic carbon (POC) and stable isotopic composition measurements, 30–70 mL of samples were filtered through pre-combusted 0.7 μm nominal pore-size glass fiber filters (Whatman GF/F, Maidstone, UK). The filter was processed, subsampled, and encapsulated as described in Remize et al. (2020) [34]. POC concentrations of all samples were measured using a CE Elantech NC2100 (ThermoScientific, Lakewood, NJ, USA) according to protocol by Remize et al. (2020) [34]. Bulk carbon isotopic composition (^13^C-POC) was analyzed by continuous flow on an Elemental Analyzer (EA, Flash 2000; Thermo Scientific, Bremen, Germany) coupled to a Delta V+ isotope ratio mass spectrometer (Thermo Scientific). Calibration was performed with international standards and in-house standard described in Table 2.

### 4.5. DIC Concentration and Bulk Carbon Isotopic Composition

Samples for dissolved inorganic carbon (DIC) concentration and stable isotopic composition were collected from the filtrate of POC samples and processed as described in Remize et al. (2020) [34]. Analyses were conducted in a gas bench coupled to a Delta Plus mass spectrometer from Thermo Fisher Scientific, Bremen, Germany (GB-IRMS).

### 4.6. Isotopic Data Processing

We used the atomic proportion of ^13^C in percent (%atom of ^13^C) to express the results instead of the δ notation due to ^13^C-labelling. Conversion between δ notation and%atom^13^C notation can be done as follow [65]:(1)%atomC13=100×(δC131000+1)×(C13C12)VPDB1+(δC131000+1)×(C13C12)VPDB
where (^13^C/^12^C)_PDB_ = 0.0112372, the ratio of ^13^C to ^12^C in the international reference VPDB standard.

Atomic enrichment (AE) of POC and DIC is then calculated from atom%^13^C-POC correction by POC_control_ values (i.e., corrected by 1.08%) and from atom%^13^C DIC corrected by control values (DIC_control_ = 1.12%), respectively, according to the following equations:(2)AEPOC=%atomC13−POCcontrol
(3)AEDIC=%atomC13−DICcontrol

### 4.7. Fatty Acids Analysis

#### 4.7.1. Fatty Acids Analysis by Gas Chromatography Flame Ionisation Detector (GC-FID)

Lipid extraction, separation of neutral and polar lipid fractions, and transesterification processes are described elsewhere [34]. Fatty acids methyl esters (FAME) samples were analyzed by gas chromatography on a Varian CP8400 gas chromatograph (Agilent, Santa Clara, CA, USA) and separated concomitantly on two columns: one polar (ZB-WAX: 30 mm × 0.25 mm ID × 0.2 µm, Phenomenex, Torrance CA, USA) and the other apolar (ZB-5HT: 30 m × 0.25 mm ID × 0.2 µm, Phenomenex, Torrance CA, USA). The FAME of *T. lutea* were quantified using C23:0 as an internal standard (2.3 µg in each lipid fraction prior transmethylation) and were identified by comparison of their retention times with commercial standards (Supelco 37 component FAME mix, the PUFA No. 1 and No. 3 and the Bacterial Acid Methyl Esther Mix from Sigma-Aldrich, Darmstadt, Germany) and in-house standards mixtures. FA concentrations were reported as µg C·L^−1^ and as % of total fatty acids from each lipid fraction. Thirty two fatty acids (FA) were thus identified and quantified: iso15:0, anteiso15:0, 14:0, 15:0, 16:0, 18:0, 22:0, 24:0, 14:1n-5, 16:1n-9, 16:1n-7, 17:1n-1, 18:1n-9, 18:1n-7, 16:2n-7, 16:2n-4, 16:4n-3, 18:2n-6, 18:3n-6, 18:3n-3, 18:4n-3, 18:5n-3, 20:2n-6, 20:3n-6, 20:4n-6, 20:4n-3, 20:5n-3, 22:2n-6, 22:4n-6, 22:5n-6, 22:5n-3, and 22:6n-3. Individual fatty acid and total fatty acid concentrations (as the sum of both fractions, named thereafter TFA) obtained in µg·L^−1^ by GC-FID were also expressed in µmolC·L^−1^ (µg·L^−1^/molecular weight of individual fatty acid × carbon number of individual fatty acid) to ease the comparison with POC concentrations expressed in µmolC·L^−1^ as well.

#### 4.7.2. Fatty Acid Compound-Specific Isotope Analysis and Processing

Samples for compound-specific isotope analyses (CSIA) of FAME were performed on a Thermo Fisher Scientific GC ISOLINK TRACE ULTRA (Bremen, Germany) using the same apolar column as mentioned above for FAME analysis. Only the fatty acids with the highest concentrations, as measured by GC-FID analyses, were considered for CSIA (namely 14:0, 16:0, 18:0, 18:1n-9, 18:2n-6, 18:3n-3, 18:4n-3, 18:5n-3, 20:5n-3, 22:5n-6, 22:5n-3, and 22:6n-3). The other FA presenting a too low signal amplitude (<800 mV) on the GC-c-IRMS did not allow precise isotope ratio analysis. Additionally, 18:1n-9 and 18:3n-3 co-eluted for GC-c-IRMS on the apolar column, but most of the isotopic signature for neutral lipids (NL) is attributed to 18:1n-9. However, in the polar fraction (PL), 18:1n-9 and 18:3n-3 are in relatively similar proportion and so were considered together. Additionally, ^13^C enrichment of 18:5n-3 could only be measured in the polar lipid fraction, but its concentration was too low in neutral lipid fraction to measure its isotope composition.

To evidence FA conversion of fatty acid A into fatty acid B in *T. lutea*, we calculated the AE_FA_ ratio (R) of product B over expected precursor A. R was defined with a confidence interval calculated at α = 0.1 (defined arbitrarily) as follows:(4)$$R=\frac{{\mathrm{AE}}_{\mathrm{FA}\left(B\right)}}{{\mathrm{AE}}_{\mathrm{FA}\left(A\right)}}$$
where A is the fatty acid hypothesized to be a precursor to fatty acid B, and AE_FA(A)_ and AE_FA(B)_ are their respective atomic enrichments at each sampling time.

If the AE of the product (B) exceeds the AE of the reactant (A), ratio > 1, then it is necessary to consider another formation process for B, since any molecule formed from A would have the same AE as A or below. If the ratio is <1, transformation of A into B is considered possible. If the ratio is close to 1, the fatty acids A and B are at equilibrium in terms of label incorporated, implying B is then synthesized simultaneously or very rapidly from A.

### 4.8. Identification within the in Silico Proteome of T. lutea of PKS Enzymes Involved PUFA Synthesis

The in silico proteome generated from last annotated version of the genome of *T. lutea* was used to identify putative proteins involved in n-3 PUFA PKS pathways [66]. We used the PKS previously identified in the haptophyte *Chrysochromulina tobin* as query for a BLASTp analysis, using e-value <10^−3^ as threshold [67]. Analysis of conserved domain was performed using the NCBI CD database V3,18 with e-value <10^−2^ as threshold. The genome location of genes encoding selected proteins was identified to evaluate genes’ proximity and occurrence of gene clusters. Proteins and cluster of proteins containing the four domains ketoacyl reductase (KR), dehydrase/dehydrogenase (DH), enoyl reductase (ER), and polyketide synthase (KS) were selected as candidates.

### 4.9. Statistical Analysis

To assess the potential effect of time and difference between balloons during algae development and of ^13^CO_2_ incorporation, Bartlett tests and ANOVA were performed on physiological and biochemical parameters, as well as PERMANOVA analysis on FA percentage separately in NL and PL. All statistical analyses were performed using R software.

## 5. Conclusions

The synthesis of long-chain PUFA in *Tisochrysis lutea* appeared to involve multiple pathways (Figure 8, Figure 9 and Figure 10). First, the assumption of the use of PKS pathway for 22:5n-6 (DPA-6) was attested regarding the fast enrichment observed for this FA as well as the absence of detectable intermediates more or equally enriched. PKS pathway appeared to be particularly efficient in *T. lutea* and induced a strong incorporation of the ^13^C-marker. However, the possibility of use of the conventional n-6 PUFA pathway should not be excluded, as 18:2n-6 presented a similar level of enrichment as 22:5n-6. It would only endorse that following desaturation and elongation steps to form 22:5n-6 were particularly dynamic and thus did not allow the accumulation of the ^13^C-label into n-6 intermediates. Within n-3 PUFA pathway, the Δ6-desaturase route seemed slower than the n-6 pathway in *T. lutea* in producing the two C_18_ polyunsaturated fatty acids 18:4n-3 and 18:5n-3. We assumed 18:4n-3 and 18:5n-3 were unlikely synthesis intermediates of 20:5n-3 and 22:6n-3, as their enrichments were lower than the latter. Although 22:6n-3 was present in higher proportion than 22:5n-6, it was not enriched as fast, possibly because its synthesis may be more complex. Indeed, 22:6n-3 could be synthesized by *Tisochrysis lutea* via a combination multiples pathway: from 22:5n-6 via ω-3 desaturase pathway, from desaturation and elongation of 20:5n-3 and 22:5n-3, and via PKS pathway. Further studies are needed to better constrain the plausible routes taken by this prymnesiophyte to produce long chain PUFA.

## Figures and Tables

**Figure 1 marinedrugs-20-00022-f001:**
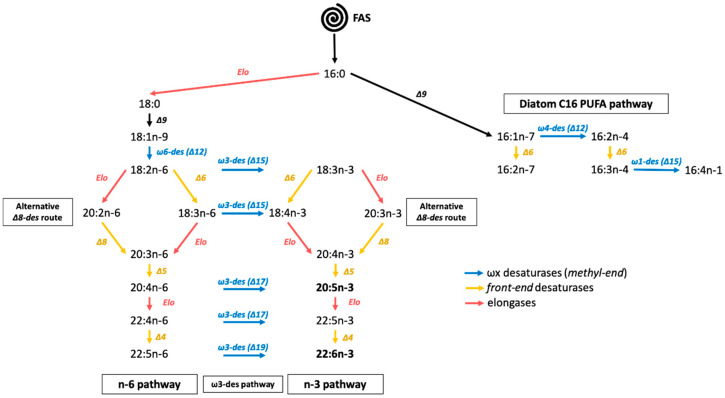
Microalgae fatty acid synthesis pathways. Desaturases are noted with “ΔX” (yellow arrows) and “ωY-des (ΔX)” (blue arrows), where X refers to the location of carbon holding the newly formed double bond from the front end (or carboxyl end) and Y its position from the methyl end. Elo: elongase, FAS: fatty acid synthase.

**Figure 2 marinedrugs-20-00022-f002:**
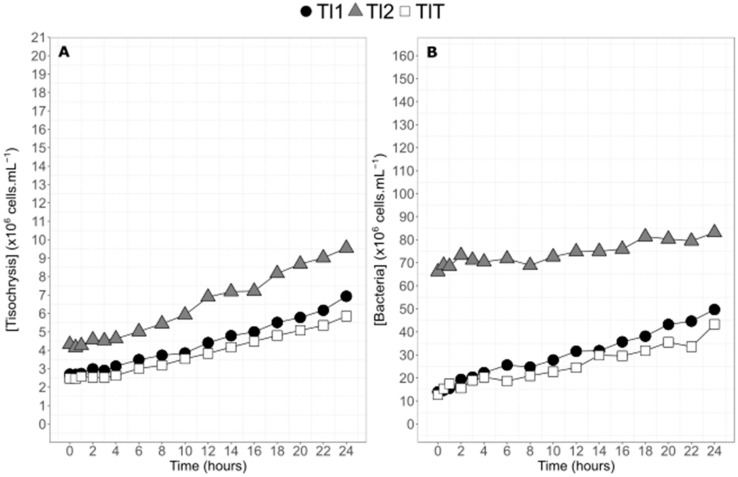
Temporal dynamics of cell concentrations of the two enriched balloons of *Tisochrysis lutea* (TI1 and TI2, filled black circles and filled gray triangles, respectively) and of the control balloon (TIT, empty squares) (**A**) and corresponding bacteria concentrations (**B**) during the 24 h experiment.

**Figure 3 marinedrugs-20-00022-f003:**
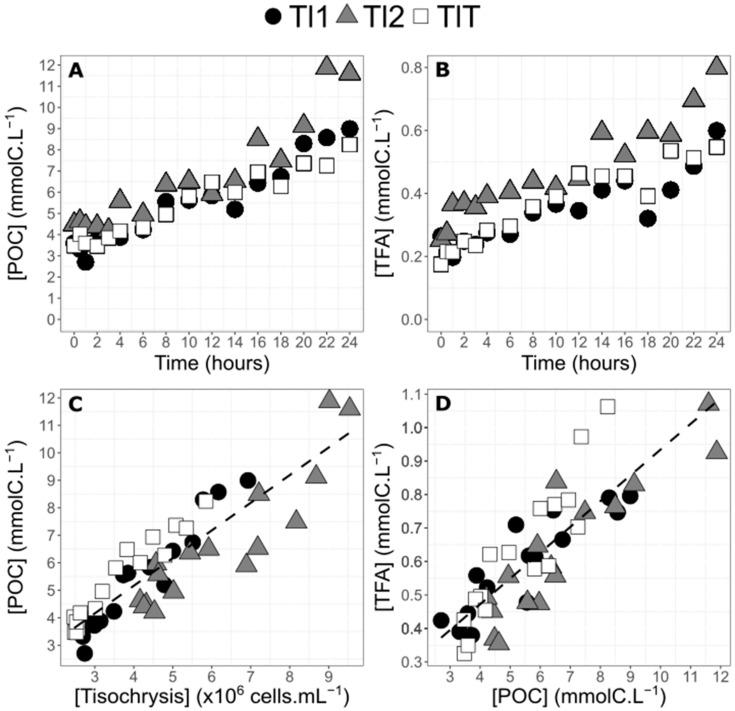
Particulate organic carbon (POC) concentration (**A**) of the two enriched balloons of *Tisochrysis lutea* (TI1 and TI2, filled black circles and filled gray triangles, respectively) and of the control balloon (TIT, empty squares) and total fatty acids (TFA) concentration according to culture age in hours (**B**). POC concentration according to algae concentration (**C**). Relation between total fatty acids and POC concentration (**D**). All regressions (dotted lines) have been calculated with data from the three balloons combined together.

**Figure 4 marinedrugs-20-00022-f004:**
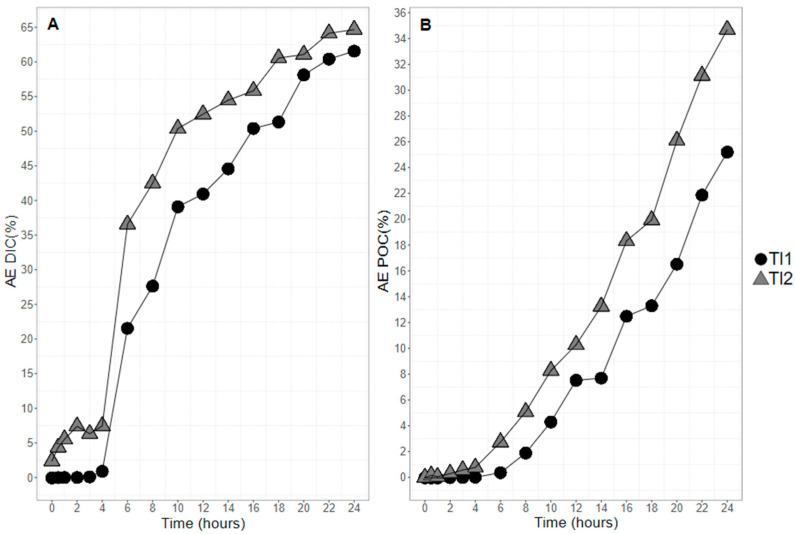
Atomic enrichment of the dissolved inorganic carbon (DIC) (**A**) and particulate organic carbon (POC) (**B**) of the two enriched balloons of *Tisochrysis lutea* (TI1 and TI2, filled black circles and filled gray triangles, respectively).

**Figure 5 marinedrugs-20-00022-f005:**
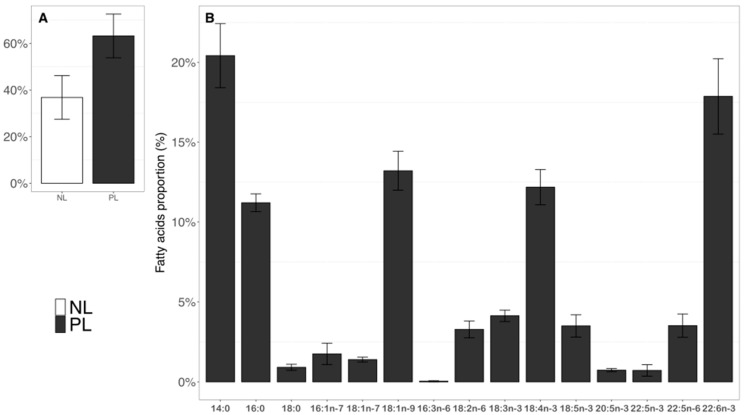
Proportions (%) of NL vs. PL (**A**) and proportions (%) of fifteen fatty acids in the PL fraction in average over the 24 h (**B**).

**Figure 6 marinedrugs-20-00022-f006:**
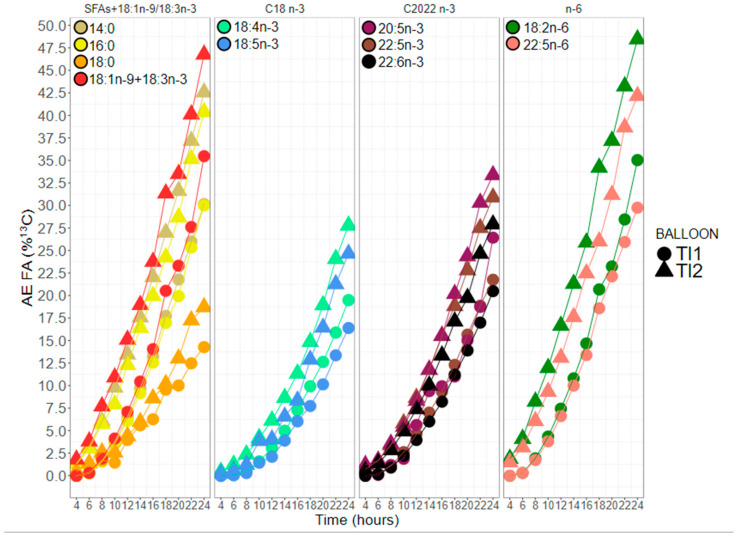
Atomic enrichment of 11 main fatty acids in the polar lipid (PL) fraction during a 24 h ^13^C labelling experiment of the two enriched balloons of *Tisochrysis lutea* (TI1 and TI2, filled circles and filled triangles, respectively).

**Figure 7 marinedrugs-20-00022-f007:**
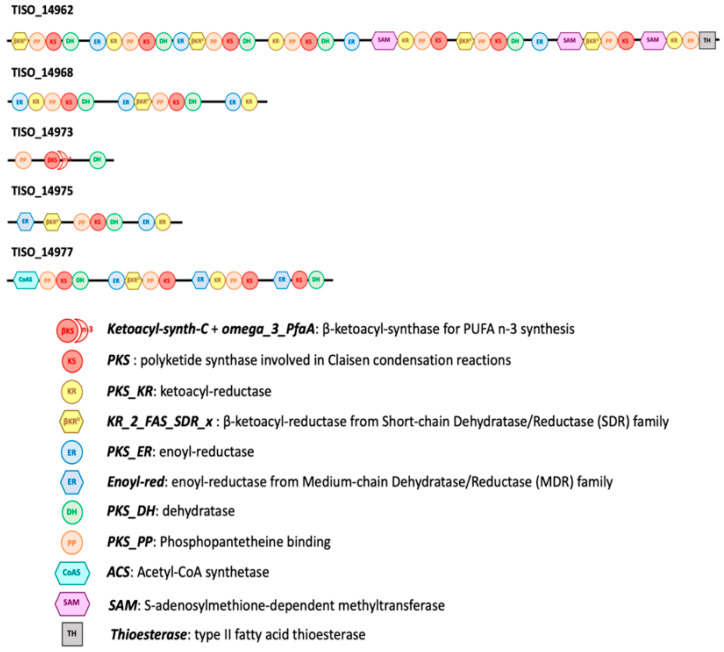
Cluster of candidate proteins suspected involved in PKS PUFA synthesis pathway in *T. lutea*. The name of each protein is annotated with TISO_ (for *Tisochrysis lutea*) and associated number. In the legend, the text written in bold italic correspond to domain names as shown in NCBI conserved domain database, followed by its suspected role.

**Figure 8 marinedrugs-20-00022-f008:**
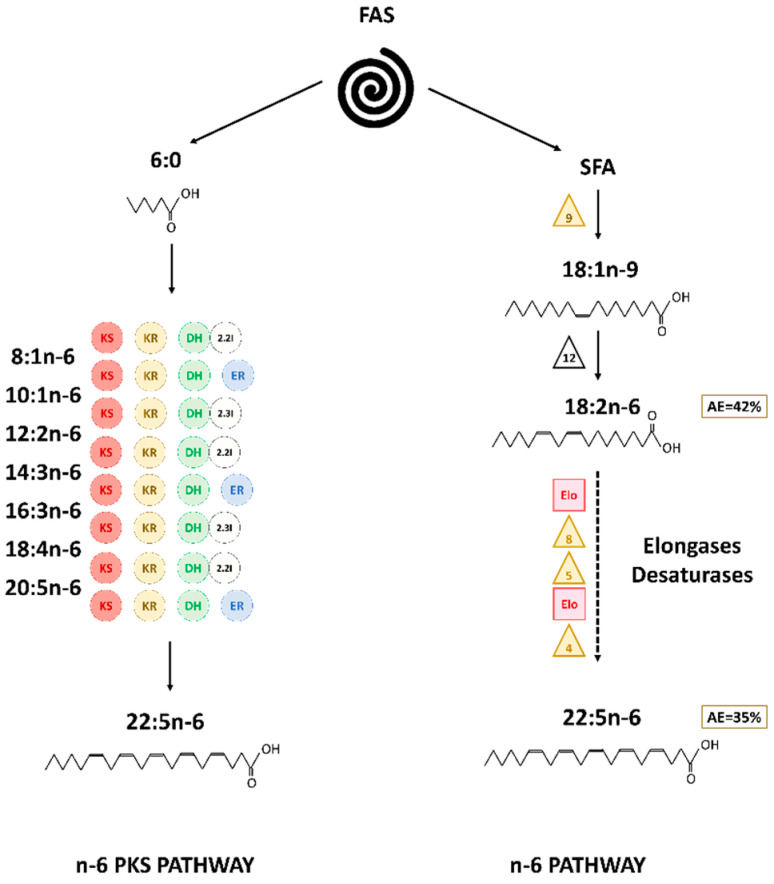
Hypothesized pathways for 22:5n-6 synthesis in *T. lutea* in the PL. Numbers in the boxes correspond to final AE value. The triangles symbolize the desaturases (front-end in yellow and methyl-end in purple), the circles the enzymes involved PKS pathway (KR: 3-ketoacyl synthase, KS: 3-ketoacyl-ACP-reductase, DH: dehydrase, 2.2I: 2-trans, 2-cis isomerase, 2.3I: 2-trans, 2-cis isomerase, ER: enoyl reductase), and the squares the elongases.

**Figure 9 marinedrugs-20-00022-f009:**
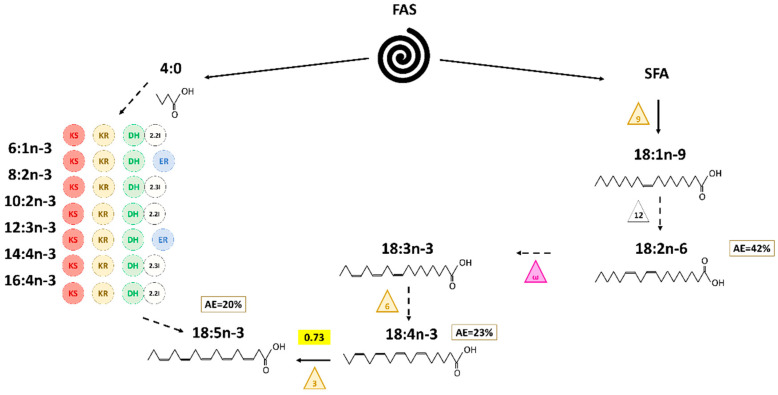
Hypothesized pathways to produce 18:5n-3 in *T. lutea*. Numbers in boxes correspond to final mean AE value, and number in the yellow box the mean value of ratio of the two surrounding fatty acids. The triangles symbolize the desaturases (front-end in yellow and methyl-end in purple), the circles the enzymes involved PKS pathway (KR: 3-ketoacyl synthase, KS: 3-ketoacyl-ACP-reductase, DH: dehydrase, 2.2I: 2-trans, 2-cis isomerase, 2.3I: 2-trans, 2-cis isomerase, ER: enoyl reductase). The directions with dashed arrows cannot be proven with the enrichment dynamics.

**Figure 10 marinedrugs-20-00022-f010:**
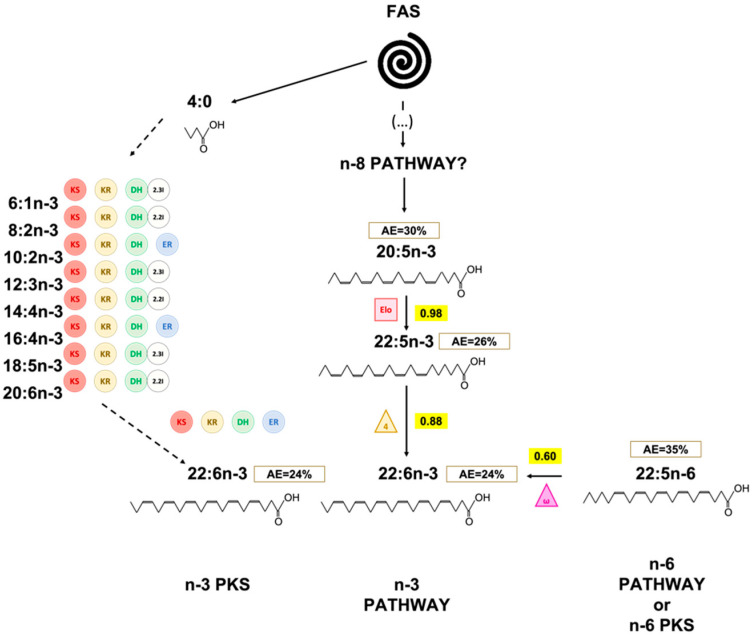
Hypothesized pathways to produce DHA in *T. lutea*. Numbers in boxes correspond to final mean AE value, and number in the yellow box is the mean value of ratio of the two surrounding fatty acids. The triangles symbolize the desaturases (front-end in yellow and methyl-end in purple), the circles the enzymes involved in PKS pathway (KR: 3-ketoacyl synthase, KS: 3-ketoacyl-ACP-reductase, DH: dehydrase, 2.2I/2.3I: 2-trans, 2-cis or 2-trans, 3-cis isomerases, ER: enoyl-reductase), and the squares the elongases.

**Table 1 marinedrugs-20-00022-t001:** Mean ratio of atomic enrichment (AE) for pairs of FA (FA_A_ vs. FA_B_) in the polar lipids (PL) (mean ± SD, *n* = 9 sampling dates t_8_ to t_24_) for the two enriched balloons (Tl1, Tl2, Tl = *Tisochrysis lutea*).

	Polar Lipids			
	Tl1		Tl2	
Fatty Acid B/Fatty Acid A	Mean *	SD	Mean *	SD
18:5n-3/18:4n-3	0.78	0.10	0.78	0.14
20:5n-3/18:5n-3	2.00	0.92	1.77	0.51
22:5n-3/20:5n-3	0.98	0.20	0.97	0.06
22:6n-3/22:5n-3	0.88	0.03	0.87	0.02
22:6n-3/20:5n-3	0.86	0.17	0.84	0.03
22:6n-3/22:5n-6	0.61	0.05	0.59	0.06

* If the AE of the product (B) exceeds the AE of the reactant (A), ratio > 1, then it is necessary to consider another formation process for B. If the ratio is <1, transformation of A into B is considered possible. If the ratio is close to 1, the fatty acids A and B are at the equilibrium in terms of label incorporated, implying B is then synthesized simultaneously or very rapidly from A.

**Table 2 marinedrugs-20-00022-t002:** List of international and in-house standards used for EA-IRMS and GB-IRMS analysis.

Description	Nature	Analysis	δ^13^C (‰)	SD
IAEA-CH_6_	Sucrose (C_12_H_22_O_11_)	13C-POC	−10.45	0.03
IAEA-600	Caffeine (C_8_H_10_N_4_O_2_)	13C-POC	−27.77	0.04
Acetanilide	Acetanilide (C_8_H_9_NO)	13C-POC	+29.53	0.01
CA21 (in-house std)	Calcium carbonate (CaCO_3_)	13C-DIC	+1.476	
Na_2_CO_3_ (in-house std)	Sodium carbonate	13C-DIC	−6.8805	
NaHCO_3_ (in-house std)	Sodium bicarbonate	13C-DIC	−5.9325

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
