# Peer review of "A 13CO2 Enrichment Experiment to Study the Synthesis Pathways of Polyunsaturated Fatty Acids of the Haptophyte Tisochrysis lutea"

_marinedrugs, 2021, doi:10.3390/md20010022_

Round 1

Reviewer 1 Report

Abstract –

See my comments about the Conclusion below.

Intro:

This reminded me that I don’t have the regular elongation/desaturation pathway solidly in mind and I had to consult a figure in another paper. I suggest including such a figure somewhere in this manuscript, with the addition of the delta 8 desaturation path. Even in a Supplementary file would be useful.

Line 40: ‘marine vertebrates’ would be more accurate statement. There’s more and more evidence that many invertebrates can synthesize EPA and DHA.

I’ll note here the language is slightly distracting. I suggest the authors find someone whose first language is English to edit it. The errors are subtle but are distracting.

Material and Methods

Line 102: Here and everywhere. Check your units. Should always have a space between the number and the unit (2 L, not 2L). In the next line, your ‘-2’ needs a superscript in m-2.s-1.

Line 116: What’s FCM? It doesn’t appear again in the manuscript so no need to abbreviate it.

Line 117: I’d just refer to it as bulk isotopic analysis to distinguish it immediately from CSIA. You do call it bulk analysis on line 141 so I’d start here.

Equation 1: I’m assuming this is correct but I’d urge the authors to confirm. A reference to a paper about sled dogs as an application of this doesn’t give me the greatest of confidence.

Line 161: I don’t know what this means “1.08% close of the values in natural marine environment”. I think you’re just correcting for the natural abundance of 13C but I’m not sure. An explicit statement would be useful.

Line 171: ‘using simultaneously two separations’. Not sure what’s meant here. Were the FAME separated on two different columns (so generating 2 chromatograms), or two columns in sequence (one resulting chromatogram)? Why use the non-polar column at all? You know you’re not going to get adequate resolution of FAME with that. Or is it that you had to use the non-polar column for CSIA but had the wax column for FID? Seems so since you have percent data for 18:1 and 18:3 but don’t have d13C values for those. More explanation is needed.

Line 177: How did you quantify? With an internal standard? What one?

Lines 193-194: Yes, I would imagine 18:1 and 18:3 might coelute on the non-polar column, but they should be well resolved on the wax column. Why not just use the data from that?

Lines 200-205: Description of AR ratio. More description and justification is needed here. For instance, the authors say that if the ratio is above 1, then FA B (the numerator) can’t be a product of FA A (the denominator). But what if the path to synthesize A from some precursor is very slow, but the conversion of A to B very fast? That should also create an R>1. Similarly, if R is near 1, you can’t know that the two FA are or are not related because their amounts would depend on the rate of their formation, not just the amount of 13C incorporated.

OK, just coming back here after looking at Table 2’s footnote. It would prevent a lot of confusion if the same information was added to the text here.

Results-

The authors refer to Figure A and B throughout without giving a Figure number…..

Figure 1. There are three symbols in this plot but only 2 included in the legend. I assume the grey is one of the enriched cultures???

Lines 250-253: Some interesting error messages here… What is missing?

Lines 287-289: Bacteria and branched FA. Which are you referring to and where is the data?

Line 296: What Figure? Presumably 5 but none is specified. Same 2 lines later. And again on Line 309 in Section 3.4. I’m not noting any more of these. Seems to be happening throughout. Some proofreading is clearly necessary.

Figure 5: the colour scheme here makes it difficult to interpret this figure. Distinguishing between cultures with just the colouring of the symbol line is not a good idea. Perhaps different shapes that are filled and empty for the two cultures? Colours as an additional sign to distinguish between the two could also be used but I don’t think you can rely on it alone.

Line 323: What do you mean by threshold value?

Discussion -

Line 374-377: Avoid 1 sentence paragraphs. Surely this can be included in another.

Line 387-390: Yes, highly active enzymes could prevent you from detecting the intermediates. I think this is exactly the problem with this ratio of AE approach and it should be expanded on.

Lines 391-404: This is an entire paragraph of supposition. Please re-write. Do you have data to support an alternative PKS pathway? If so, make that clear. If not, say you don’t and remove Figure 7 and this paragraph.

Line 406: Doesn’t 18:5 show the lowest enrichment? And since you couldn’t resolve 18:1 and 18:3, any discussion of 18:4 being synthesized from 18:3 seems pointless since you can’t determine d13C of 18:3. I suggest removing this paragraph entirely.

Figure 8: Same comments. You can’t include an AE here for 18:3n-3. You can’t assume that it’s the same as the co-eluting peak. It is really unfortunate that you were not able to resolve 18:3 from 18:1. I would be happier to see the supposition surround Fig 7 stay than this. I would hate for anyone to think that a d13C value from co-eluting peaks is reliable. Continuing to include Fig 8 and surrounding discussion about potential synthesis pathways of 18:4n-3 will only confuse readers.

Line 412-420: This is interesting. More of this and less of supposition about data you don’t have would make for a stronger discussion.

Line 421-422: Did you intend this to start the next paragraph? Seems to make more sense there. Or do you mean desaturation of 18:4n-3?

Line 423-428: Couldn’t it also be possible that 18:5 is from the regular pathway and 22:6 from the PKS? Or vice versa? Is the PKS pathway associated with particular organelles? Do we know? And why haven’t you included a Figure to describe this (like you did for 18:4n-3)? You have strong data for this. Why not make the most of it?

Line 446-450: Might be good to remind us here that Qi et al (and you) found a similar result for n-6 FA (lack of intermediates linked to delta 8 pathway).

Overall, the Discussion could be stronger. I encourage the authors to focus on the conclusions that they can reasonably draw from their data (for instance, routes to synthesize of 22:6) and downplay the supposition (paths for 22:6n-6, 18:4n-3).

Reviewer 2 Report

This manuscript addresses the synthesis pathways of PUFAs in the haptophyte Tisochrysis lutea using 13C labelling method monitering 13C-labelled CO2 incorporation into FAs. The manuscript provides an update on n-3 PUFA synthesis pathway that addition to the common n-3PUFA pathway from 18:3n-3 to 22:6n-3, there are contribution of the omega-3 desaturase activity which can convert 22:5n-6 into 22:6n-3. The authors further suggest the use of the PKS pathway from 20:5n-3. I think this paper is of wide interest for the readers of this journal.

Minor points

At the line 250-253 and 488, There are some inappropriate descriptions references “Error!...”.

Numbers of Figures and table are missing throughout the manuscript.

Round 2

Reviewer 1 Report

These responses are reasonable.